# Mechanical Fault Diagnosis of a DC Motor Utilizing United Variational Mode Decomposition, SampEn, and Random Forest-SPRINT Algorithm Classifiers

**DOI:** 10.3390/e21050470

**Published:** 2019-05-06

**Authors:** Zijian Guo, Mingliang Liu, Huabin Qin, Bing Li

**Affiliations:** 1HLJ Province Key Lab of Senior-Education for Electronic Engineering, Heilongjiang University, Harbin 150080, China; 2Key Laboratory of Information Fusion Estimation and Detection, Heilongjiang Province, Harbin 150080, China

**Keywords:** variational mode decomposition, sample entropy, random forest, SPRINT algorithm

## Abstract

Traditional fault diagnosis methods of DC (direct current) motors require establishing accurate mathematical models, effective state and parameter estimations, and appropriate statistical decision-making methods. However, these preconditions considerably limit traditional motor fault diagnosis methods. To address this issue, a new mechanical fault diagnosis method was proposed. Firstly, the vibration signals of motors were collected by the designed acquisition system. Subsequently, variational mode decomposition (VMD) was adopted to decompose the signal into a series of intrinsic mode functions and extract the characteristics of the vibration signals based on sample entropy. Finally, a united random forest improvement based on a SPRINT algorithm was employed to identify vibration signals of rotating machinery, and each branch tree was trained by applying different bootstrap sample sets. As the results reveal, the proposed fault diagnosis method is featured with good generalization performance, as the recognition rate of samples is more than 90%. Compared with the traditional neural network, data-heavy parameter optimization processes are avoided in this method. Therefore, the VMD-SampEn-RF-based method proposed in this paper performs well in fault diagnosis of DC motors, providing new ideas for future fault diagnoses of rotating machinery.

## 1. Introduction

Reliable operation of DC motors is crucial to factories, which has a huge impact not only on personal safety but also on efficiency of operations [1]. Therefore, it is of great significance to study the fault diagnosis of motors. As vibration signals generated by mechanical components often provide various dynamic information about the status of the mechanical system, many studies on DC motor fault diagnosis have focused on mechanical vibration signals [2,3], which have achieved successful application to a certain extent in the past decades.

The most commonly employed method of fault diagnosis combines signal processing techniques with the vibration signals of rotating machinery to extract fault features. Vibration signals obtained from rotating machinery are periodic and continuous. Thus, time-frequency analysis is the optimal signal process [4,5,6]. In recent years, various methods, such as wavelet decomposition (WD), wavelet packet decomposition (WPD), empirical mode decomposition (EMD), and local mean decomposition (LMD), have been extensively applied in the field of motor fault diagnosis. However, wavelet decomposition and wavelet packet decomposition are nonadaptive signal analysis methods, which need to select the wavelet basis function in advance [7,8]. The eigenmode function of EMD will be submerged in the background of hostile noise, which makes it impossible to extract physical characteristic parameters [9,10]. As an improved method, ensemble empirical mode decomposition (EEMD) has effectively solved these problems. To provide a unified reference framework in the time-frequency domain, EEMD adds white noise of finite (rather than infinitesimal) amplitude in the original vibration signals [11], but the central section of the algorithm is still the EMD method, which still has endpoint effects and inaccurate calculation problems. EWT (empirical wavelet transform) and LMD are two adaptive signal decomposition methods that are based on a recursive mode [12,13,14]. However, LMD and EWT algorithms are essentially adaptive signal decomposition methods based on a recursive mode, with endpoint effects and modal aliasing problems [15,16]. In recent years, another important method for diagnosis-based signals is based on principal component analysis (PCA) of the data. Santos-Ruiz et al. utilized dynamic PCA and an extended Kalman filter (EKF) to describe, detect, and quantify fluid leaks. The paper firstly proposed a data-driven system based on PCA for describing, detecting, and quantifying fluid leaks in the experimental pipelines [17]. Compared with the two approaches of steady-state estimation and EKF, Santos-Ruiz et al. proposed a method for leak detection and isolation (LDI) in pipelines based on data fusion [18]. The two methods ensure normal operation of the pipelines by on-line monitoring of tubing flow through modeling. However, the working environment of the motor is changeable, making it difficult to model accurately. The two methods both need a lot of data, but this experiment was simulated to monitor the early fault of the motor and could not produce a lot of fault data. Thus, the two methods mentioned above are not suitable for early fault diagnosis of the motor. Variational mode decomposition (VMD) is a novel and adaptive signal decomposition method proposed by Dragomiretskiy and Zosso in 2014 [19]. As a nonrecursive variational mode signal decomposition method, it can decompose complex multicomponent signals into different modes. Compared with former methods, it has obvious advantages in signal decomposition accuracy and noise robustness. Meanwhile, mode aliasing can also be solved [20].

When the operational state of rotating machinery changes, its dynamic characteristics tend to show nonlinearity and complexity, and its vibration signals also present nonlinearity and nonstationarity [21,22]. Traditional nonlinear dynamic theories [23], such as fractal dimension [24,25] and Lyapunov exponent [26], have been employed in the field of fault diagnosis for many years. However, these methods require a large amount of data in parameter estimation, while early fault data of motors are generally scarce. Entropy is used to describe the overall statistical characteristics of the signal, which is strongly generalizable to the overall characteristics of the signal. Jaynes proved in 1957 that thermodynamic entropy was identical with information-theory entropy of the probability distribution [27]. Based on the research, Basaran and Yan presented a damage metric based on the second law of thermodynamics and statistical mechanics [28]; verification of the damage model has been performed by extensive comparisons with laboratory test data of low cycle fatigue. In 2010, Naderi postulated that the thermodynamic entropy of metals, undergoing repeated cyclic loads and reaching the point of fracture, is a constant that is independent of geometry, load, and frequency. Its generation has applications in determining the fatigue life of components that undergo cyclic bending, torsion, and tension–compression [29]. Furthermore, Sosnovskiy made an attempt to propose a generalized theory of evolution based on the concept of tribo-fatigue entropy. Meanwhile, a mechanothermodynamic function was constructed for the specific case of fatigue damage of materials as a result of temperature variation from 3 K to 80% of the melting temperature based on the analysis of 136 experimental results [30]. According to the probability of new patterns generated in signal time series, approximate entropy (ApEn) was proposed [31] by Pincus, which measured the signal from the aspect of measuring the complexity of the time series. ApEn of the vibration signal describes the complexity and irregularity of the signal. The ApEn variation of motor equipment under different operating conditions can characterize the operation status of the machinery during this period. As an improved algorithm based on ApEn, sample entropy (SampEn) can lower the deviation introduced by approximation entropy into self-matching entropy. Therefore, this paper employed the sampEn algorithm to extract feature signals [32].

The random forest (RF) algorithm was proposed by Breiman in 2001 [33]. RF is composed of multiple decision trees [34], which is similar to the bagging algorithm [35]. Based on the bootstrap method [36], it resamples and generates multiple training sets, and the final results are determined by the voting principle of multiple decision trees. Different from conventional algorithms, it is a method that randomly selects and splits attribute sets. Compared with LQV (learning vector quantization) [37], ELM (extreme learning machine) [38], and the decision tree, it has the advantages of a fast training speed, powerful approximation ability, and high simulation accuracy. However, when there are unrecognizable faults, such a fault diagnosis method based on traditional machine learning is still unable to help engineers find where the fault is, which reduces the efficiency of fault diagnosis [39]. To solve the above problems, a new fault diagnosis method based on the RF algorithm is proposed in this paper. Through improving the bagging method of decision trees, this method employs a conditional probability index to split unbiased nodes of the decision tree, and it synthesizes the classification results of the decision tree according to weighted voting principles.

As indicated above, the main contribution of this paper is to propose a novel mechanical fault diagnosis method for DC motors based on VMD, SampEn, and RF, during which the optimal RF classifier is established through the SPRINT algorithm. Firstly, the signal acquisition system was designed in LabVIEW 2014, and then the acquired signals were de-noised by the translation invariant of the wavelet. After extracting the physically meaningful modes by the VMD algorithm, the signal characteristics are calculated by the SampEn method. Finally, the improved RF classifier is employed to identify the labeled samples.

## 2. Variational Mode Decomposition (VMD)

### 2.1. A Brief Description of VMD

VMD extracts different vibration signals from different vibration statuses by continuously updating eigenmode functions and utilizing an alternating direction multiplier method to constantly update various intrinsic mode functions and central frequencies [20,21]. Subsequently, it decomposes the self-adaptive quasi-orthogonal signal f into a discrete quasi-orthogonal band-limited signal $N_{k}$ with certain sparsity, and each mode revolves around the center frequency $\omega_{t}$. The estimated K central angular frequencies $\omega_{t}$ are obtained by setting the finite broadband parameter (a) and the central angular frequency initialization method. Furthermore, the modal functions $U_{k}$ are obtained according to different central angular frequencies. Each modal function is a single component AM-FM function [40]:

The band-limited intrinsic mode function (BIMF) is defined as:(1)$$U_{k}=A_{k}\left(t\right)\cos\left[\phi \left(t\right)\right].$$

In the formula, the phase function ϕ(t) is a non-monotonic decreasing function, $\phi {\left(t\right)}^{'}>0$. The envelope $A_{k}\left(t\right)>0$, and the change rate of $A_{k}\left(t\right)$ as well as the instantaneous angular frequency $A_{k}\left(t\right)>0$ is low, which is much slower than the phase function ϕ(t).

According to Carson’s rule, the BIMF prior estimate [41] is:(2)$$BIMF_{AM-FM}=2\left(f_{AM}+f_{PM}+\Delta f\right).$$
where Δ*f* is maximum deviation of homeopathic frequency, $f_{FM}$ refers to offset rate of instantaneous frequency, $\mathrm{and}\;f_{PM}$ means the highest frequency of envelope $A_{k}\left(t\right)$. In order to estimate the frequency bandwidth of IMF (Intrinsic Mode Function) components, the following steps must be taken:
For each modal function $U_{K}$, the marginal spectrum is obtained by a Hilbert transformation.The exponent-modified method is utilized to move the spectrum of modal functions to their estimated central frequencies.The bandwidth of each modal function is obtained by Gauss smoothing.

The process of VMD can be considered as the construction and solution of a constrained variational problem, as described by Equation (3):(3)$$\begin{array}{c} min\\ \left\{u_{k}\right\},\left\{\omega_{k}\right\} \end{array}\left\{\sum_{k=1}^{K}{\left\|\partial_{t}\left\{\left[\sigma \left(t\right)+\frac{j}{\pi t}\right]*u_{k}\left(t\right)\right\}e^{-j\omega_{k}t}\right\|}_{2}^{2}\right\},$$
wherein $u_{k}$ denotes the subsignals, $\;\omega_{k}$ refers to the center frequencies of the submodes, $\partial_{t}$ indicates the partial derivative of the function to find time *t*, σ(t) means the unit pulse function, *j* indicates the imaginary unit, and * is the convolution. In order to render the problem unconstrained, a quadratic penalty term and Lagrangian multipliers are introduced, and the problem is renewed to Equation (4).
(4)$$L\left(u_{k},\omega_{t},\lambda \right)=\alpha \overset{k}{\underset{k=1}{\sum }}\partial {\left\|\left[\left(\sigma \left(t\right)+\frac{j}{\pi t}\right)\times u_{k}\left(t\right)e^{-j\omega_{k}t}\right]\right\|}_{2}^{2}+{\left\|f\left(t\right)-\overset{K}{\underset{k=1}{\sum }}u_{k}\right\|}_{2}^{2}+\left(\lambda \left(t\right),f\left(t\right)\overset{K}{\underset{k=1}{\sum }}u_{k}\left(t\right)\right)\;\;\;,$$
where α  is a penalty term (called equilibrium constraint parameter) and λ is a Lagrange multiplicator. The saddle point is found by using the alternate direction method of multipliers (ADMM) [42].

The decomposition mode number K is determined artificially, and the frequency-domain expression of mode ${\hat{u}}_{k}^{1}$, the center frequency of each mode function $\omega_{k}^{1}$, and the largrangian multiplicator are initialized. Meanwhile, the mode $u_{k}$ and the center frequency $\omega_{k}$ are updated separately by Equations (5) and (6):(5)$${\hat{u}}_{k}{}^{n+1}=\frac{\hat{f}-\sum_{i<k}{\hat{u}}_{i}{}^{n+1}-\sum_{i>k}{\hat{u}}_{i}{}^{n}+\frac{{\hat{\lambda }}^{n}}{2}}{1+2\alpha {\left(\omega -\omega_{n}^{k}\right)}^{2}};$$
(6)$$\omega =\frac{\int_{0}^{\infty }\omega {\left|{\hat{u}}_{k}{}^{n+1}\left(\omega \right)\right|}^{2}d\omega }{\int_{0}^{\infty }{\left|{\hat{u}}_{k}{}^{n+1}\right|}^{2}d\omega }\;.$$

The modes $u_{k}$ and the center frequencies $\omega_{k}$ can be obtained through each updated equation. The largrangian multiplier is renewed by the following equation:(7)$${\hat{\lambda }}^{n+1}={\hat{\lambda }}^{n}+\tau \left(\hat{f}\left(\omega \right)-\sum_{k}{\hat{u}}_{k}{}^{n+1}\left(\omega \right)\right)\;\;.$$

The iteration is updated until it converges to Equation (8).
(8)$$\sum_{k}\frac{{\left\|{\hat{u}}_{k}{}^{n+1}-{\hat{u}}_{k}{}^{n}\right\|}_{2}^{2}}{{\left\|{\hat{u}}_{k}{}^{n}\right\|}_{2}^{2}}<\varepsilon$$

It can be concluded that this algorithm needs to set K, α,τ, and the tolerance convergence criterion in advance. Compared with the former two parameters, τ and the tolerance convergence criterion barely have an impact on the decomposed signal. Hence, the default value was adopted in this paper. Prior experience was required to get K. If the parameter was set without ascertaining, it would be difficult to ensure the accuracy and efficiency of signal decomposition. Therefore, searching the best parameters is key to the method, which promotes the following research.

### 2.2. VMD Algorithm Simulation

The simulated signal was composed of three types of subsets, namely three different harmonics, $x\left(t\right)=x_{1}+x_{2}+x_{3}$, wherein $\;x_{1}=\cos\left(6*\pi *t\right),x_{2}=1/9\sin\left(52*\pi *t\right),\mathrm{and}\;x_{3}=1/27\cos(600*\pi *t)$ [43]. The simulation signal x(t) is decomposed by VMD. The simulation signal and its decomposition diagram are shown in Figure 1.

The three modal components (presupposition K = 3) obtained from the decomposing simulation were signals u1, u2, and u3, which corresponded to the input signals x1, x2 and x3, respectively. Both magnitude and frequency of the modal component and original signal matched well with each other, as shown Figure 1. Furthermore, the spectrum distribution of the input signal is shown in Figure 2, and the obtained spectrum distribution of each modal signal is shown in Figure 3. It can be observed that the three sums of modal frequency after decomposition can perfectly coincide with the frequency of the total decomposition signals in the frequency range. Compared with the original signal, results showed that the three decomposed modal frequencies can coincide with the original signals in the frequency range.

### 2.3. Study of the Parameter Decomposition Modes of the VMD

#### Simulation Experimental Results

Periodic signals were employed to verify the effect of the VMD modal value on signal decomposition. The mode number was set as K = 2, 3, 4, and 5, and the penalty factor was set as *a* = 2000. As shown in Figure 4 below, when the mode number was set as K = 2, the decomposed signal frequencies of 4 and 26 Hz were superimposed, which caused modal mixing. Meanwhile, as shown in Figure 5 and Figure 6 where $u_{1}\left(t\right),\mathrm{and}\;u_{2}\left(t\right)$ represent the cosines of frequencies 4 and 26 Hz, respectively, if the K value was higher, a new condition, called false mode, would appear. False mode $u_{3}\left(t\right)$ was rendered when K = 4, while false modes of $u_{3}\left(t\right),\;u_{5}\left(t\right)$ were rendered when K = 5. By tracing different K values, different central frequency curves can be obtained. As is shown in the Figure 7, when K = 3, the effects of the simulation signal will be better than the others, proving the right choice of parameters. 

The variation in central frequency of different K values is shown in Table 1. It can be observed that no sign of similar modal or modal mixing phenomenon appeared, when K = 3. But when K = 2, decomposition of periodic signals led to modal mixing. Moreover, false modes appeared in the process of decomposition as the value of K grew. 

## 3. Sample Entropy 

**For** a sequence of vectors of dimension m, $X_{m}\left(1\right)⇢X_{m}\left(N\;-\;m\;-\;1\right)$
**do**
The distance ${d[X}_{m}\left(i\right){,X}_{m}\left(j)\right]$ between vector $X_{m}\left(i\right)$ and $X_{m}\left(j\right)$ is defined as the absolute value of the maximum difference between the two corresponding elements [44,45]
$$d\left[X_{m}\left(i\right),X_{m}\left(j\right)\right]=max_{k=0,⇢,m-1}\left(\left|x\left(i+k\right)-x\left(j+k\right)\right|\right)$$As for $X_{m}\left(i\right)$*,* the number of *j*
(1 ≤ j ≤ N − m, j ≠ i) whose distance between $X_{m}\left(i\right)$ and $X_{m}\left(j\right)$ is less than or equal to *r* is counted and denoted as Bi. For 1 ≤ i ≤ N−m, there is:(9)$$B_{i}^{m}=\frac{1}{N-m-1}*B_{i}\;\;$$Define $B^{\left(m\right)}\left(r\right)$ as
(10)$$B^{\left(m\right)}\left(r\right)=\frac{1}{N-m}\overset{N-m}{\underset{i=1}{\sum }}B_{i}^{\left(m\right)}\left(r\right)$$

**For** the dimension raised to *m* + 1, the number of the distance between $X_{m+1}\left(j\right)$ and $X_{m+1}\left(j\right)\;(1\;<\;j<\;N-m,\;j\neq \;i)$ that is less than or equal to *r* is calculated and denoted as $A_{i}$. $A_{i}^{m}\left(r\right)$ is defined as:(11)$$A_{i}^{m}\left(r\right)=\frac{1}{N-m-1}A_{i}\;\;$$

Define $A^{m}\left(r\right)$ as
(12)$$A^{m}\left(r\right)=\frac{1}{N-m}\overset{N-m}{\underset{i=1}{\sum }}A_{i}^{m}\left(r\right)\;\;\;\;\;\;$$


**End For**


$B^{m}\left(r\right)$ is the probability of two sequences to match *m* points under similar tolerance *r*, while $A^{m}\;\left(r\right)$ is the probability of two sequences matching *m* + 1 points [26]. 

**If**N→∞, sample entropy is defined
(13)$$\mathrm{SampEn}\left(m,r\right)=\underset{N\to \infty }{\lim}\left\{-ln\left[\frac{A^{m}\left(r\right)}{B^{m}\left(r\right)}\right]\right\}\;\;$$

**Else if***N* is a finite value, it is seriated the following Equation:(14)$$\mathrm{SampEn}\left(m,r,N\right)=-\ln\left[\frac{A^{m}\left(r\right)}{B^{m}\left(r\right)}\right]$$


**End For**


It can be observed that the value of sample entropy was related to the values of *m* and *r*. Therefore, it was important to determine the values of the two parameters to calculate sample entropy. Based on our research, when *m* = 1 or 2, or *r*= 0.25 Std (Std is the standard deviation of initial data x(i),i=1,2→N), the obtained sample entropy had the desired statistical properties. In this research *m* = 2 and *r* = 0.2 Std.

## 4. Random Forest

### 4.1. Bootstrap Resampling Algorithm

Bootstrap resampling is an integrated learning algorithm to sample the original data, which is an important component of the random forest algorithm. The algorithm requires extracting a sample from different samples $\left\{x_{1},x_{2},\to x_{n}\right\}$ in set *S*. A new set $S^{*}$ is formed, and the probability of excluding a sample $x_{i}\left(i=1,2,\to ,n\right)$ in set $S^{*}$ is [35]:(15)$$p={\left(1-\frac{1}{n}\right)}^{n}.$$

When →∞,
(16)$$\underset{n\to \infty }{\lim}p=\underset{n\to \infty }{\lim}{\left(1-\frac{1}{n}\right)}^{n}=e^{-1}\approx 0.368\;\;\;\;.$$

Although the total number of samples in the new set $S^{*}$ is equal to the original set *S* (the total is *n*), the new set $S^{*}$ may contain duplicate samples. After removing the samples, it contains only about 1 − 0.368 × 100% = 63.2% samples in the original set *S*.

### 4.2. Bagging Algorithm

A bootstrap algorithm is the earliest learning strategy, and the main idea is shown in Figure 8. The specific steps can be described as follows [36]:Produce T training sets randomly through the bootstrap method of resampling;Each training set is used to generate the corresponding decision tree;As for the test set sample X, each decision tree is applied to test the corresponding categories;Choose the category that has the most outcomes from the decision tree for testing sample X through voting.

### 4.3. Decision Tree

Decision tree learning is an inductive learning method based on specific examples. Generally, the recursive method of "from top to bottom, divide and rule" is employed to divide the search space into several nonoverlapping subsets. The schematic diagram of the decision tree is shown in Figure 9. Each non-leaf node represents the input attribute of the training set data, and leaf node represents the value of the target category attribute. "yes" and "no" represent the positive examples and counterexamples in the instance set [34].

The purpose of decision tree learning is to generate trees with strong generalization abilities. There are many categories of decision trees, such as ID3, C4.5, and SPRINT. Although C4.5 overcomes the shortcomings of ID3 [46], it still has problems such as complex branches, large scales, and low efficiencies of the decision tree.

The SPRINT algorithm is a scalable, parallel, and inductive decision tree, which operates fast and allows multiple processors to create a collaborative decision tree model. Compared with the above two algorithms, it can better build a decision tree in parallel, making it suitable for larger data sets [33]. The SPRINT algorithm is based on the Gini index [47]:(17)$$\mathrm{gini}\left(T\right)=1-\sum_{j=1}^{n}p_{j}^{2}\;,$$
where $p_{j}$ is the frequency of occurrence of class *j*. If set T is divided into two parts, T1 and T2, corresponding to M1 and M2 respectively, then the Gini index of this division is:(18)$${\mathrm{gini}}_{\mathrm{division}}\frac{m_{1}}{m}\mathrm{gini}\left(T_{1}\right)+\frac{m_{2}}{m}\mathrm{gini}\left(T_{2}\right).$$

The minimum Gini value is selected as the standard of segmentation. If it is a numerical range or continuous field, the possible segmented points are the midpoint of the two values. As for a discrete field, they are all subsets of attribute values. 

### 4.4. Random Forest Algorithm

The traditional machine learning model, like a neural network, is accurate in its prediction. However, the learning speed of a neural network is slow, and simple problems need to be trained hundreds of thousands of times, which will easily fall into local minimums. Compared with a neural network, random forest can effectively balance the error when there is imbalance in classification with high-speed network training. Based on the above statements, a RF approach based on SPRINT was proposed in this paper.

Similar to the bagging algorithm, this method was based on a bootstrap method to resample and generate multiple training sets. Unlike the bagging algorithm, RF chooses split attribute sets randomly when reconstructing a decision tree. The detailed random forest algorithm flow is shown in the Figure 10 (Assuming the number of attributes of the sample is *M*, *m* is an integer greater than zero and less than *M*):Samples are collected by the bootstrap method, and training sets $S_{1},S_{2},S_{3}\to S_{T}$ will be generated randomly;Each training set is employed to generate the corresponding decision tree $C_{1},C_{2},C_{3}\to C_{T}$;Every tree grows intact without pruning;As for the test set sample X, it gets the corresponding category by testing each decision number;The category with the most output from T decision tree is taken as the category of test set sample X by voting.

### 4.5. Random Forest-United SPRINT Algorithm

Given the classifiers $h_{1}\left(x\right),h_{2}\left(x\right),\to h_{k}\left(x\right)$ and random vector Y, X, the edge function is defined as:(19)$$mg\left(X,Y\right)=av_{k}I\left(h_{k}\left(x\right)=Y\right)-max_{j\neq Y}av_{k}I\left(h_{k}\left(x\right)=j\right).$$

In the formula, *I*( ) is an indicative function, $av_{k}\left(\to \right)$ is the average value, and, thus, the generalization error of the classifier is defined as:
$$PE^{*}=P_{x,y}\left(mg\left(x,y\right)\right)$$

The above conclusions are extended to the random forest $h_{k}\left(X\right)=H\left(X,\theta_{k}\right)$. If the number of trees in the forest is large, the following theorems can be obtained by the law of large numbers and the structure of trees:

**Theorem** **1.**
*With the increase of the number of trees, all random vectors*
$\theta \to PE^{*}$
*trends*
(20)$$P_{x,y}(p_{\theta }\left(h\left(x,\theta \right)=y\right)-max_{j\neq Y}p_{\theta }\left(h\left(x,\theta \right)=j\right)<0).$$


It has been proven in literature [10] that over-fitting does not occur with the increase of numbers. In the meantime, the generalization error will converge to an upper bound.

Random forest edge function:(21)$$mv\left(X,Y\right)=P_{\vartheta }\left(h\left(X,\theta \right)=Y\right)-max_{j\neq Y}P_{\vartheta }\left(h\left(X,\theta \right)=j\right).$$

Classification strength of classifier {h(X,θ)}:(22)$$S=E_{x,y}mr\left(x,y\right).$$

If S ≥ 0, according to Chebyshev inequality:(23)$$PE^{*}=var\left(mr\right)/S^{2}.$$

Var (*mr*) is required to have the following form:(24)$$\{\begin{array}{c} var\left(mr\right)=\bar{\rho }{\left(E_{\theta }sd\left(\theta \right)\right)}^{2}\\ var\left(mr\right)<\bar{\rho }\left(E_{\theta }var\left(\theta \right)\right) \end{array}.$$

In the formula, $\bar{\rho }$ is the mean value of the correlation coefficient, showing the correlation degree among trees in a random forest.
(25)$$\bar{\rho }=E_{\theta ,\theta^{\prime }}\left(\rho \left(\theta ,\theta^{\prime }\right)sd\left(\theta \right)sd\left(\theta^{\prime }\right)\right)/E_{\theta ,\theta^{\prime }}\left(sd\left(\theta \right)sd\left(\theta^{\prime }\right)\right),$$
and
(26)$$\{\begin{array}{c} E_{\theta }var\left(\theta \right)\leq E_{\theta }{\left(E_{x,y}mg\left(\theta ,x,y\right)\right)}^{2}-S^{2}\\ E_{\theta }Var\left(\theta \right)\leq 1-S^{2} \end{array}.$$

The following conclusions are received from Formulas (23), (25), and (26).

**Theorem** **2.**
*The upper bound of generalization error for random forest is defined as*
(27)$$PE^{*}\leq \bar{\rho }\left(1-S^{2}\right)/S^{2}.$$


*S* is the classification strength of trees. The upper bound of the generalization error is determined by the classification accuracy of each decision tree and the degree of correlation between trees. It increases with the degree of correlation between trees, and it is also proportional to the classification intensity of each tree.

RF employs a classification and decision tree (SPRINT) to grow a single classification tree, which is different from the traditional SPRINT tree. The specific steps of random forest generation are as follows: The random forest utilizes the bootstrap resampling method to extract 63.2% of the samples from the original training sample set to generate a subsample set, where each subsample corresponds to a classification tree. At the same time, the original samples that are not sampled are called out-of-bag data (OOB). OOB data are used to evaluate the classification accuracy of classifiers.Each sub-sample set becomes a single classification tree. At each tree node, *m* feature vectors are picked out of *M* feature vectors, according to the empirical formula $m\;=\;\mathrm{int}\sqrt{M}$. According to the principle of minimum nonpurity of nodes, feature α is chosen as the classification attribute of the node.According to the feature α, the node is divided into two branches, and then the best feature from the remaining features is searched. The classification tree can grow sufficiently and the impurity of each node can be minimized through the above method.In the classification phase, the classification label is a combination of the results of all classification trees. RF is based on the voting principle.
$$C_{p}=argmax_{c}\left(\frac{1}{N}\sum_{i=1}^{N}I\left(\frac{n_{h_{i},c}}{n_{h_{i}}}\right)\right)$$

In the formula, *N* is the number of decision trees in the forest, $n_{h_{i},c}$ is the classification result of class *C* for tree $h_{i}$, and $n_{h_{i}}$ is the number of leaf nodes of the tree.

## 5. Experimental Results and Analysis

An INV-1612 data acquisition system from the China Orient Institute of Noise & Vibration (COINV) Company was used for mechanical vibration signal acquisition. It had the following advantages: accurate measurement, high sampling rate, and a large number of ports. In this paper, the type of DC motor was a ZHS-5 multi-function rotor experiment bench. Therefore, before we collected its vibration signals, we had to consider the frequency range of the vibration signals and the sampling rate of devices. The SLM-6000 sensor signal conditioner acquired data with six test channels and an ADC resolution of 16 bits, which can extend external trigger circuits. These advantages were suitable for continuous signal acquisition in this experiment.

Besides the rotor experiment bench and signal conditioner, an LC0159 piezoelectric acceleration sensor (LANCE TECHNOLOGIES INC., Hebei, China), a DC stabilized voltage source, and an AFT-0931 signal conditioner (Electronic Technology Co., Ltd., Shijiazhuang, Hebei, China) were applied to build the vibration signal acquisition system for the DC motor. The frequency range of the LC0159 piezoelectric acceleration sensor was 1–12,000 Hz, and the range and sensitivity were 500 g and 10 mV/g, respectively, while the frequency response and gain of the AFT-0931 signal conditioner were 0.5–45 and 5 kHz, respectively.

In this paper, vibration signals were collected by the INV-1612 fault data acquisition system, as shown in Figure 11 and Figure 12. Firstly, the vibration sensor LC0159 was employed to transform the vibration signals into analog electrical signals. Secondly, the analog electrical signals were inputted into the SLM-6000 sensor signal conditioner, and the filtering and amplification processes of the analog electrical signals were then completed in the signal conditioner. In addition, the signal conditioner could prevent shock signals with big amplitudes from damaging the host computer. Finally, the analog electrical signal was inputted into the SLM-6000 for AD conversion. The design framework of the entire signal acquisition system block diagram is shown in Figure 13.

In this paper, 50 samples were collected for each mechanical state by the acquisition system, and there was a total of 200 samples. Figure 14 shows the time domain waveform of the normal signal and three types of fault signals (rotor asymmetry, load-shafting misalignment, and rotor system misalignment-rubbing coupling faults). The sampling frequency was 10 Ks/s, and the sampling time was 0.7 s. The three kinds of faults were more different from normal conditions. These differences were mainly caused by the impact or friction of each mechanical component of the operating mechanism—when the mechanical fault occurred, the impact or friction process changed to a certain extent; therefore, the instantaneous frequency, instantaneous amplitude, and occurrence time of each vibration event must be different. 

As mentioned previously, after signal acquisition work was completed, signals needed to be de-noised. Utilizing a variable time window and frequency window to analyze the signal, a wavelet transform was very suitable for vibration signal de-noising of rotating machinery based on a nonstationary wavelet threshold function. Meanwhile, the smoothness of the de-noised signal was considered. A wavelet transform method was applied in signal de-noising in this paper. It was an effective method to extract the vibration signal through the wavelet threshold de-noising method. Daubechies-9 wavelet was employed after experimental comparison. The de-noised signals of four types of original signals are shown in Figure 15.

### 5.1. Signal Decomposition and Feature Extraction

The measured signals were more complicated and different from simulation signals. Here, seven components have been decomposed from measured signals. As shown in Figure 16, each mode component was a narrow band AM-FM signal. The frequency distribution of the vibration signals changed because of the mechanical component.

The key was to determine the modal value K. When the signal was over-decomposed, the central frequency of the mode was identical. In the meantime, the correlation coefficients of each mode with different K values were compared with the original signals to verify the practicability of the method (center frequency). Based on the formula, the center frequency of the fault was calculated, as is shown in Table 2. It was proved that the center frequencies were greatly different when K = 3 and 4. When K = 8, a similar number of modalities appeared, thus, the best modal value was seven [38].

For further proof, the correlation coefficients between each mode and the original signal were obtained by formulas. In Table 3, the correlation coefficient increased at first and then decreased. However, when K = 8, it shows that over-decomposition occurred. When K = 3, 4, and 5, modal mixing appeared. The optimum mode number was obtained when K = 7, which further proved the accuracy of the center frequency method.

Modal mixing does not easily happen in the VMD algorithm. The first six envelop spectrums were compared in this paper. Figure 17 schematically illustrated the envelope spectrums of all modes that were decomposed by VMD (take Fault I for example). It can be seen from the diagram that the envelope spectrum of each component of decomposed signal can clearly reflect fault information. The envelope spectrum of decomposed signal components by EMD is shown in Figure 18. The phenomenon of modal mixing appeared in mode 4. When the number of modes was six, the envelope spectrum could not accurately reflect fault information, which proved that the decomposition signal effect of the VMD algorithm was better than that of EMD.

Envelope entropy processed the envelope signals obtained after demodulation operations into a probability distribution sequence, and its calculated entropy reflected the sparse characteristics of the original signals. If the IMF component contained more fault characteristic information, the envelope diagram would be relatively dense. As shown in Figure 17, compared with EMD, the envelope spectrum generated by VMD decomposition was very dense, which included much of the fault information of the physical system. The IMF 1 envelope spectrum indicated it was an optimum component by EMD decomposition, as shown in Figure 18. However, the IMF 1 envelope spectrum was too messy, and it was hard to extract valid fault information.

Meanwhile, the envelope spectrums of EWT, LMD, and VMD were compared. LMD and EWT were new nonstationary signal processing methods. They could adaptively decompose the signal according to the characteristics of the signal itself. They were superior to the EMD method in terms of restraining the endpoint effect, reducing iteration times, and preserving signal information integrity. However, LMD and EWT had lots of shortcomings. When the number of smoothing times was large, hysteresis occurred, and the step size could not optimally be determined in smoothing. Finally, compared with other methods, the VMD algorithm could extract all the information of the fault signal accurately in the field of signal decomposition, which ensured that modal mixing could be thoroughly solved.

### 5.2. Feature Extraction

When *M* was one, characteristics of the random forest algorithm could not be shown, for the random forest algorithm required selecting *m* eigenvectors from *M* eigenvectors. In addition, in order to prove that random forest could show better performance under relatively simple eigenvector conditions, the time domain characteristics of motor vibration signals were extracted in this paper, and the feature vectors of each signal were extracted by using sample entropy, which were denoted as ${C\;=\;[C}_{1}{,C}_{2}{,C}_{3}{,C}_{4}{,C}_{5}{,C}_{6}{,C}_{7}]$. The entropy value of samples under different fault conditions is shown in Table 4.

Firstly, sample entropy was calculated by Formulas (9)–(14) and regarded as the feature vector. At the same time, the corresponding four states were labeled D = [1,2,3,4] for the expected input of the classifier. Table 4 shows that the eigenvectors of each state were similar, and the eigenvectors of different states were distinct. Subsequently, according to the process of classifier design, 200 feature vectors were randomly divided into testing sets and training sets. For each type, 30 feature vectors were randomly selected as training samples, and 20 other feature vectors were used as testing samples. In addition, 10-fold cross-validation was employed in the paper, and 120 training samples were equally divided into two data sets again, one for training, where cross validation was performed 10 times. Besides, the loop traversal interval was [0.1, 2], and traversing step length TL was 0.1. Feature vectors were inputted into the optimal classifier (RF with “tree = 500”) for testing. Finally, the classifier was evaluated by the testing set.

### 5.3. Fault Diagnosis of the DC Motor

There are many classifiers (decision tree) in random forest. The parameters of each classifier are different, and the training sample sets of each classifier are also different. Therefore, different classifiers were generated to make the diagnosis results of each classifier diverse. Finally, final classification results were generated by voting. It effectively improved the accuracy of diagnosis.

From the figure below, the recognition rates of four conditions were demonstrated. Blue circles represented correct recognition samples, and red stars represented wrong recognition samples. The result shows that, in the 80 samples set in the testing set, the normal state was not misidentified, and there was one sample recognition error in the other three faults. The vibration signal recognition rates were: Normal 100%, Fault I 95%, Fault II 90%, and Fault III 90%. This paper also discussed the effect of random forest generalization. In Figure 19, the abscissa represented all decision trees (500 trees) of random forest and the training set samples; the ordinate represented the test set samples. The abscissa and longitudinal coordinates satisfied X + Y = 500.

If the error classification sample was close to the center of the graph (the intersection of line y = x and X + Y = 500, P (250, 250)), and the training set was the same as the decision tree of the test set in the implementation of the algorithm, the error recognition of the sample was acceptable (Figure 19b), verifying good performance of sample generalization.

Conversely, if the misclassified samples deviated from the center, as shown in Figure 19c,d, there was a gap between the decision tree of the training set and the test set in the whole algorithm process, and the samples would be misclassified. This was unreasonable and errors were not allowed, revealing that the generalization performance of the classifier was slightly poor.

Random forest contains different decision trees, which also exerts a certain impact on their generalization performance. Many experiments were conducted in this paper for detailed discussion and research. In order to reduce the impact of randomness, 100 random forest models were established after the decision tree was determined, and then the average of the accuracy was taken as the classification accuracy under the current decision tree. As shown in Figure 20, it was ideal to select 400–500 decision trees in random forest in the case of fault identification of rotating machinery, taking the speed of the decision tree and modeling in random forest into consideration.

LQV, ELM, and decision trees are widely employed classifiers in rotating machinery fault diagnosis, and they all work perfectly to a certain extent. In order to prove the ability to differentiate normal and fault types of these classification methods, the LQV, ELM, decision tree, and random forest classifiers were compared in this paper, as shown in Table 5.

The value of the hidden layer neurons played a crucial role in the prediction results of the neural network. As for the ELM classifier, the value of the hidden layer neurons was the number of samples in the training set, and the activation function of hidden layer neurons was ‘sig’. Regarding the LQV classifier, we followed the literature [3] to design the LQV into a three-layer neural network, which included an input layer, a competition layer, and a linear output layer, and the number of hidden layer neurons was 20.

The parameters of the decision tree are listed in the above Table 5, which also shows the final recognition results of these three classifiers and recognition rates for different parameters. Optimal parameters for each method can be found in this table. Because of the similar fault patterns, the recognition rates of fault II and fault III showed little difference, indicating that the vibration signals of the fault I condition were very different from the vibration signals of the other three conditions. The optimized random forest classifier was compared with LQV, ELM, and decision tree classifiers. The result revealed that with the same feature extraction method, the decision tree method was better than other two neural networks, and the recognition rate of the improved RF classifier was higher than other methods.

## 6. Conclusions

This paper presents a new method for mechanical fault diagnosis of DC motors. Firstly, the original vibration signals were collected by the designed acquisition system, which was designed by Labview2014. Secondly, VMD was employed to decompose the vibration signals into a series of physically meaningful modes. Thirdly, the SampEn method was used for feature extraction. Finally, the improved random forest method was adopted to identify four mechanical conditions, where the parameter “tree” of RF was optimized by a loop traversal method. The simulation and practical test results demonstrate the following advantages of the new method:Compared with EMD, LMD, and EWT methods, the VMD method can better decompose the signals into a series of physically meaningful modes, and it also can solve the mode aliasing well.The proposed SampEn method is more superior in feature extraction. The feature curves of different types of signals obtained by the SampEn method are more dispersed than approximate entropy method, which is good for classification.The parameter “tree” of RF needs to be determined, which usually depends on human choice. Combined with SPRINT, the loop traversal method can effectively find the best parameter of the decision tree.According to the final recognition rates, the VMD-SampEn feature extraction method and the optimal RF classifier are more suitable for mechanical fault diagnosis of DC motors. The proposed RF-SPRINT method provides a new idea for mechanical feature extraction of DC motors.

## Figures and Tables

**Figure 1 entropy-21-00470-f001:**
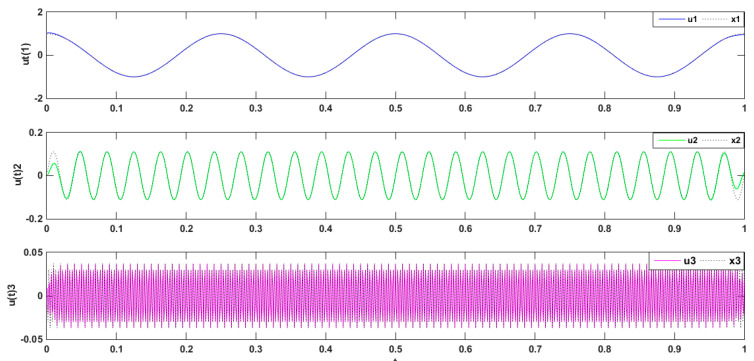
Comparison between simulated signal and modal components.

**Figure 2 entropy-21-00470-f002:**
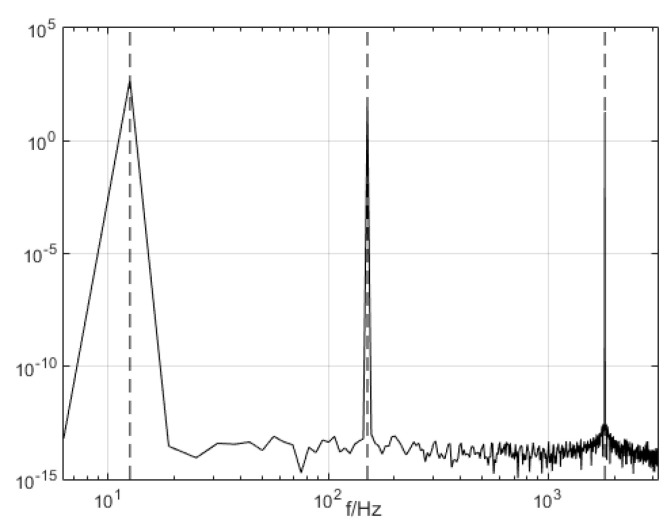
The frequency spectrum distribution of the input signal.

**Figure 3 entropy-21-00470-f003:**
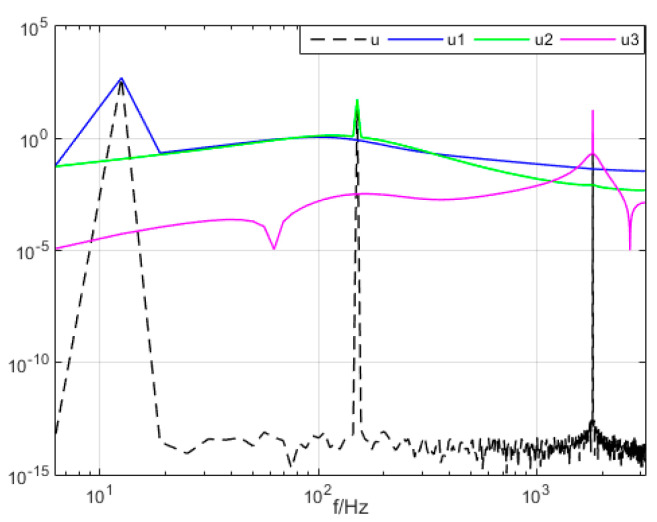
The mode signal spectrum with variational mode decomposition (VMD).

**Figure 4 entropy-21-00470-f004:**
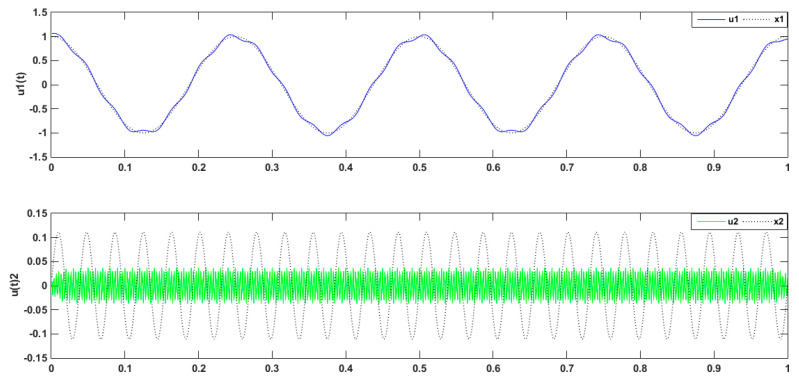
Decomposed signal by VMD (K = 2; *a* = 2000).

**Figure 5 entropy-21-00470-f005:**
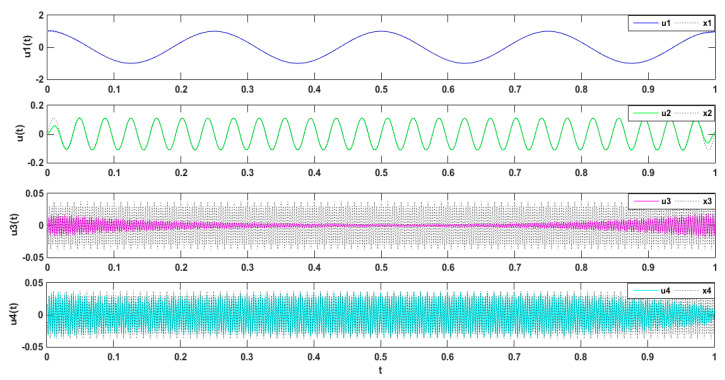
Decomposed signal by VMD (K = 4; *a* = 2000).

**Figure 6 entropy-21-00470-f006:**
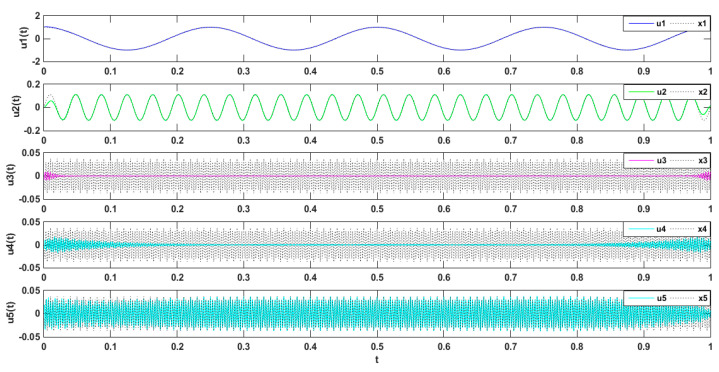
Decomposed signal by VMD (K = 5; *a* = 2000).

**Figure 7 entropy-21-00470-f007:**
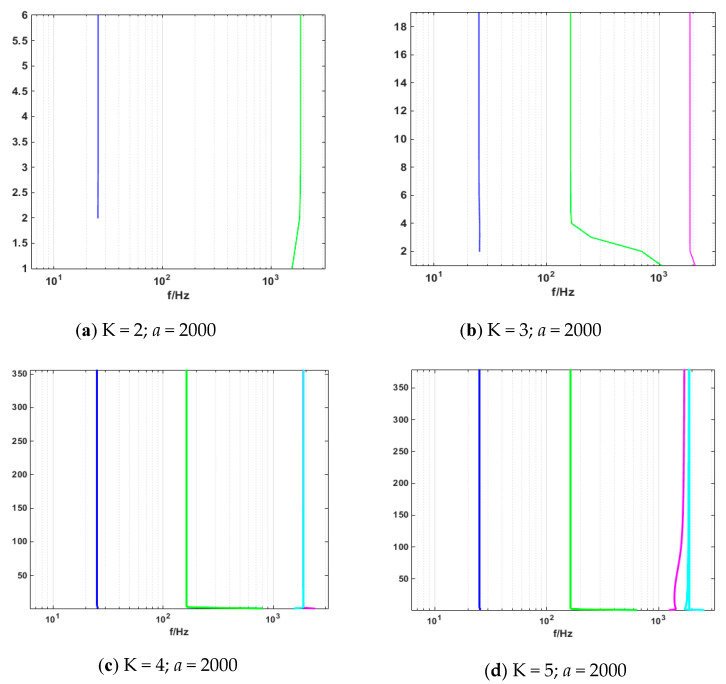
Change of center frequency.

**Figure 8 entropy-21-00470-f008:**
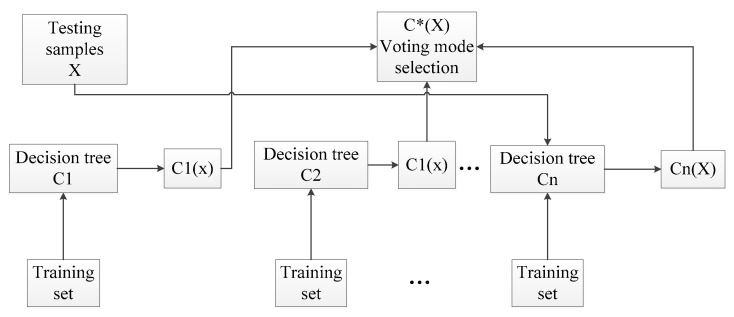
Bagging algorithm flowchart.

**Figure 9 entropy-21-00470-f009:**
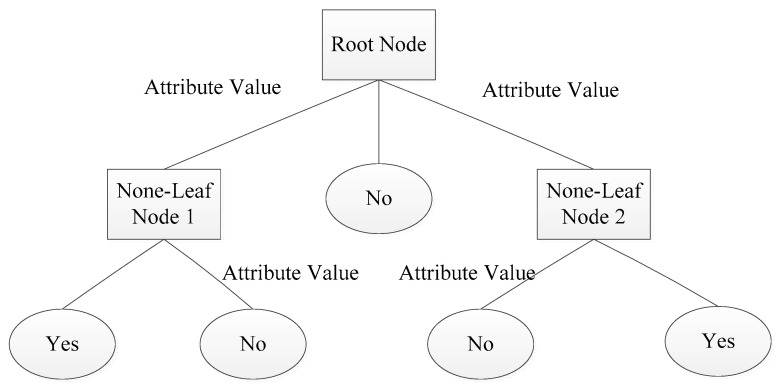
Schematic diagram of a decision tree.

**Figure 10 entropy-21-00470-f010:**
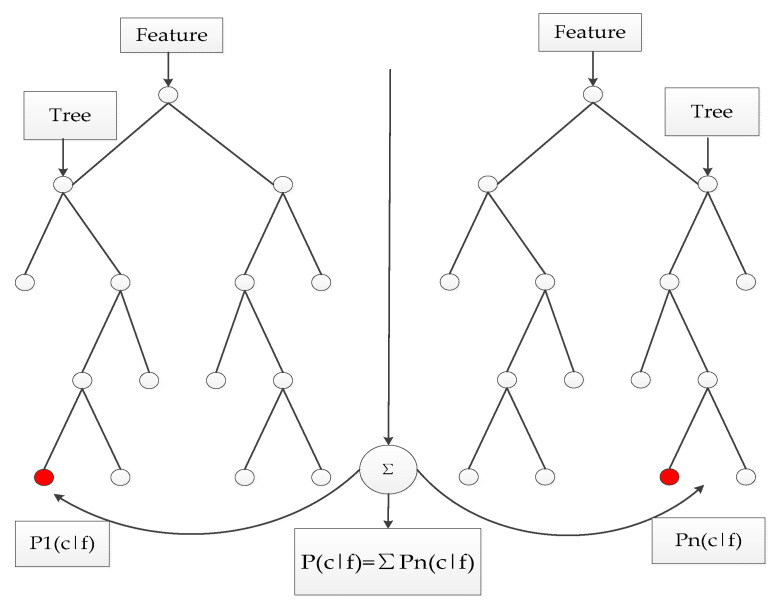
Schematic diagram of random forest.

**Figure 11 entropy-21-00470-f011:**
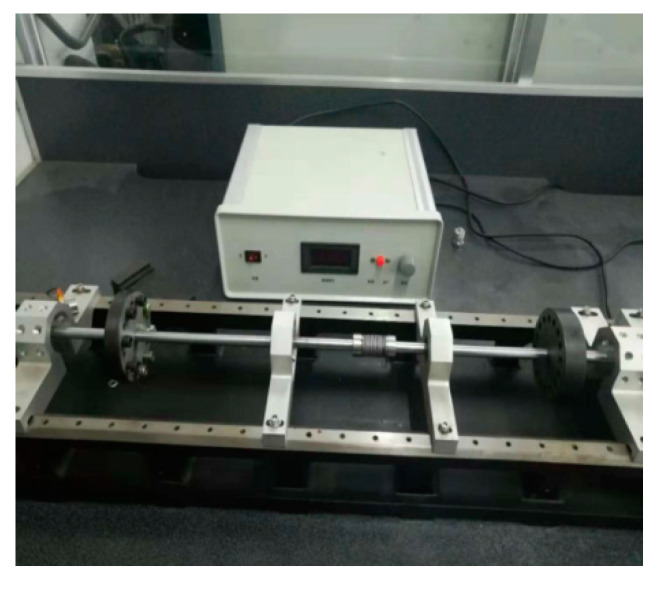
ZHS-5 multi-function rotor experiment bench.

**Figure 12 entropy-21-00470-f012:**
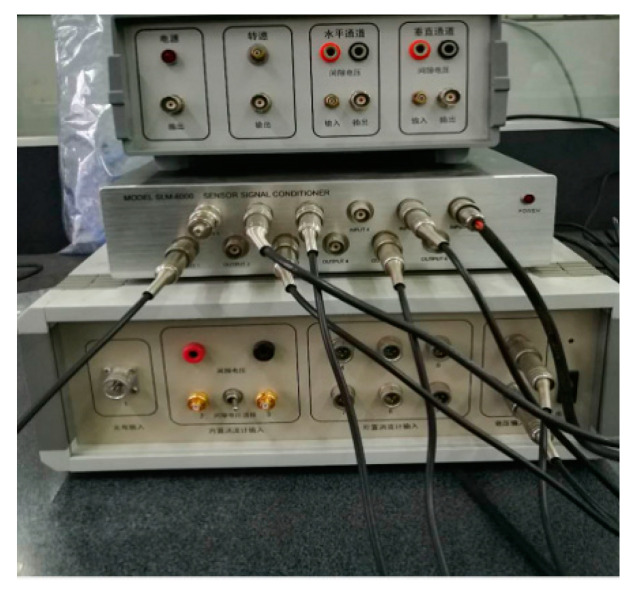
Model SLM-6000 sensor signal conditioner.

**Figure 13 entropy-21-00470-f013:**
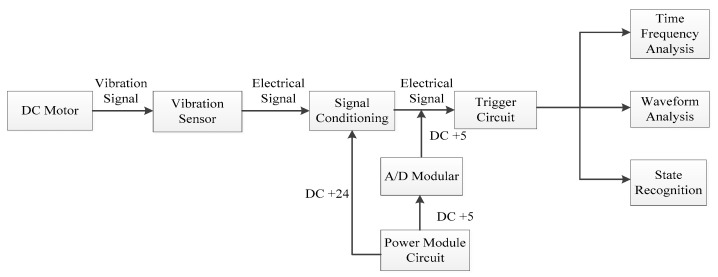
Design framework of the entire signal acquisition system.

**Figure 14 entropy-21-00470-f014:**
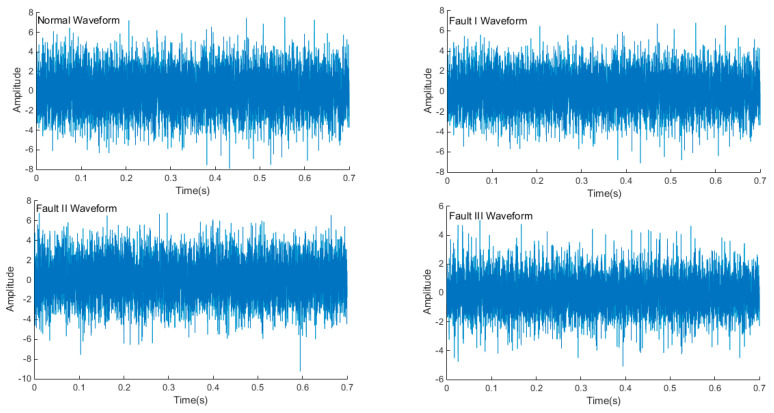
Time domain waveform of each mechanical state.

**Figure 15 entropy-21-00470-f015:**
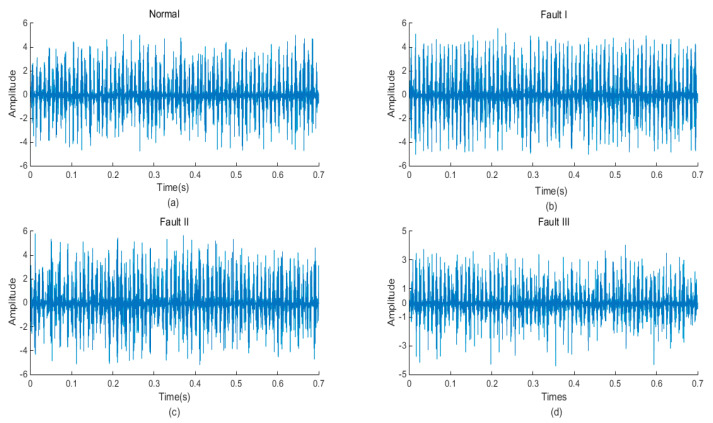
The de-noised signal normal: (**a**); Fault I: (**b**); Fault II: (**c**); Fault III: (**d**).

**Figure 16 entropy-21-00470-f016:**
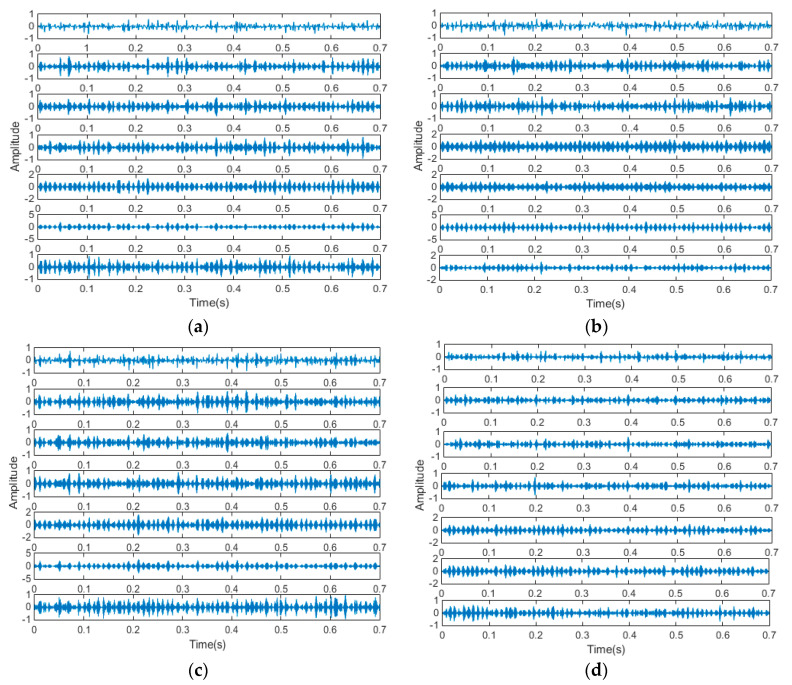
Vibration Components of the four types of signals obtained by the EWT method. (**a**) Normal type; (**b**) Fault I; (**c**) Fault II; and (**d**) Fault III.

**Figure 17 entropy-21-00470-f017:**
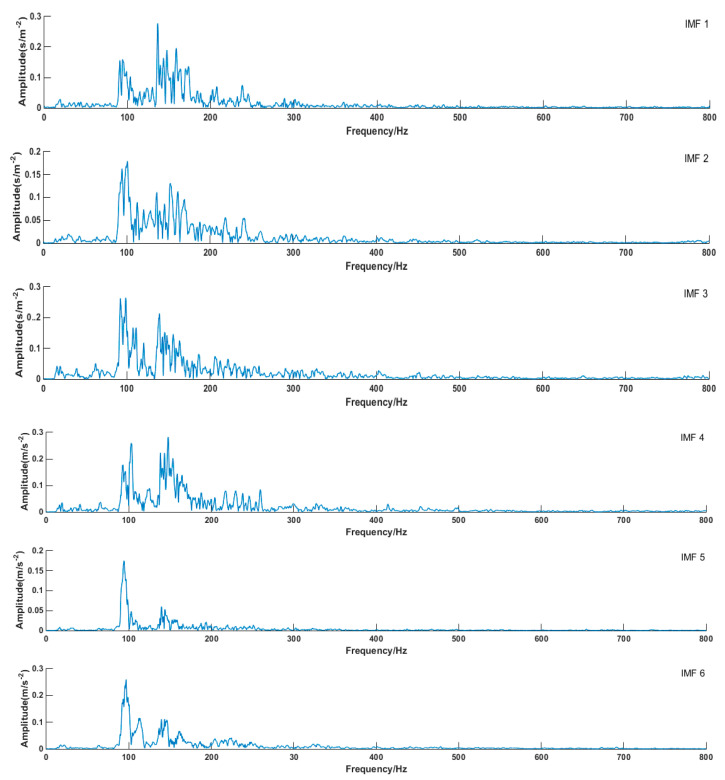
Envelope spectrum of each mode (VMD).

**Figure 18 entropy-21-00470-f018:**
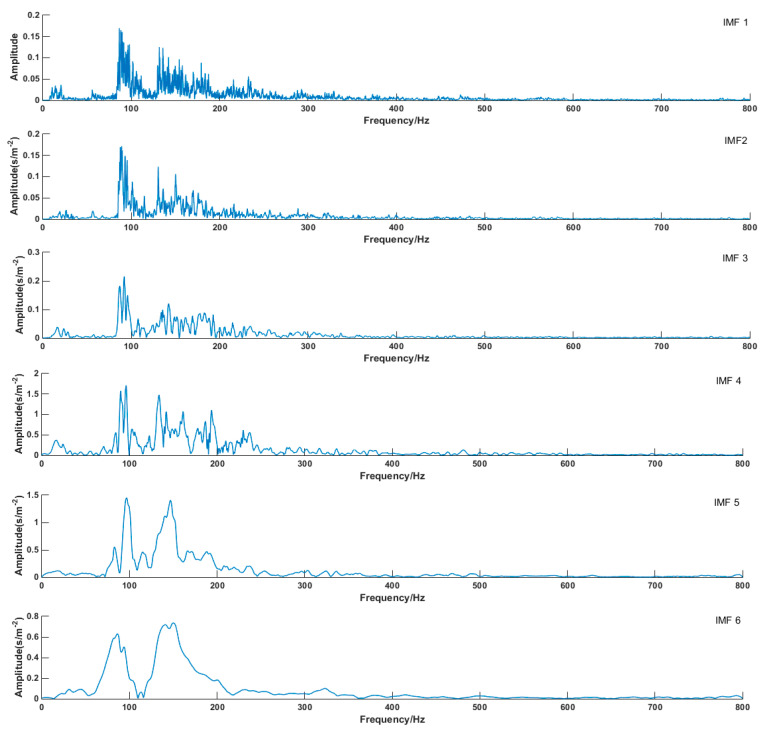
Envelope spectrum of each mode (EMD).

**Figure 19 entropy-21-00470-f019:**
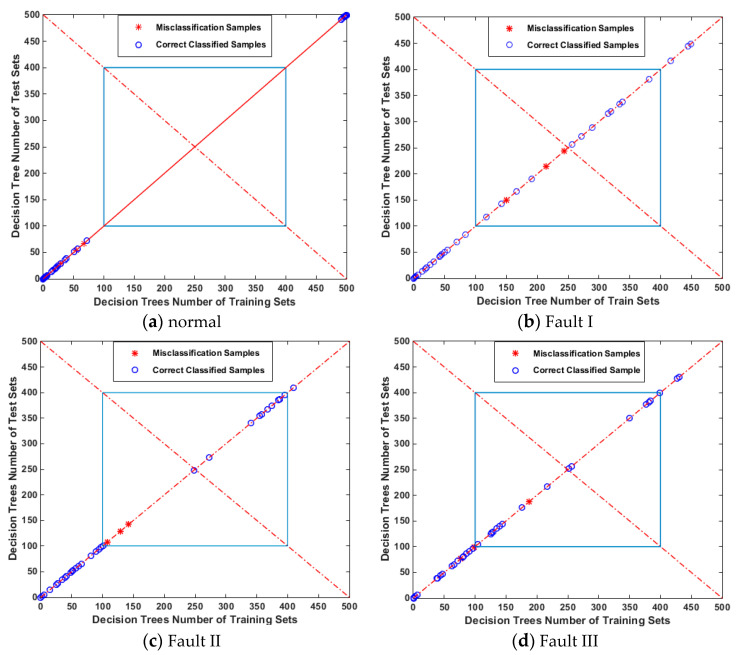
Performance Analysis of Forests Classifier.

**Figure 20 entropy-21-00470-f020:**
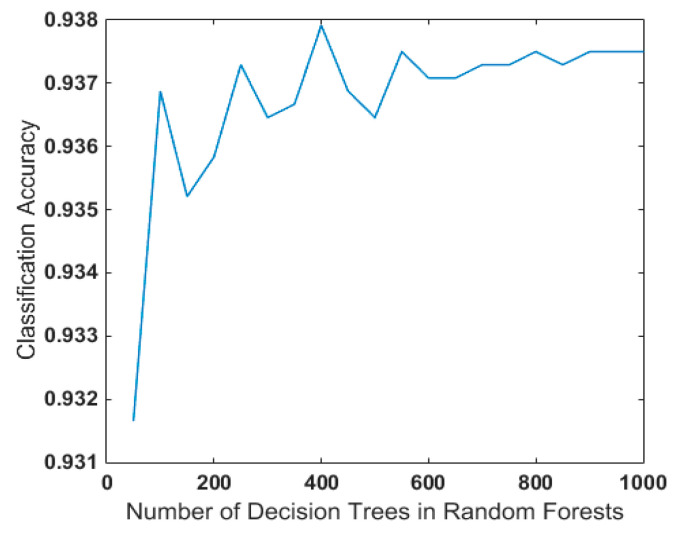
Influence of Decision Tree Number on Performance in Random Forest.

**Table 1 entropy-21-00470-t001:** Change of center frequencies in VMD under different mode values.

Mode Value	Center Frequency				
	U1	U2	U3	U4	U5
K = 2	25.76	1884.32			
K = 3	25.13	163.36	1884.96		
K = 4	25.13	163.36	1884.96	1884.96	
K = 5	25.13	163.36	1763.06	1874.27	1885.58

**Table 2 entropy-21-00470-t002:** Change of center frequencies in VMD under different K values (Fault III).

Model Value	Center Frequency							
	U1	U2	U3	U4	U5	U6	U7	U8
K = 3	95.86	456.25	554.80					
K = 4	89.69	200.74	462.75	554.80				
K = 5	87.17	198.56	372.27	464.10	554.96			
K = 6	83.56	117.50	283.25	384.21	463.85	645.28		
K = 7	83.56	117.49	283.21	382.64	464.32	553.70	645.28	
K = 8	21.67	98.64	183.31	285.25	382.27	464.32	553.70	554.88

**Table 3 entropy-21-00470-t003:** Change of Correlation Coefficients in VMD under different K values (Fault III).

Model Value	Correlation Coefficient							
	$\rho_{1}$	$\rho_{2}$	$\rho_{3}$	$\rho_{4}$	$\rho_{5}$	$\rho_{6}$	$\rho_{7}$	$\rho_{8}$
K = 3	0.3366	0.5512	0.5930					
K = 4	0.3029	0.3213	0.5485	0.5916				
K = 5	0.2945	0.3055	0.3417	0.5324	0.5876			
K = 6	0.2928	0.3029	0.3355	0.5258	0.5669	0.3570		
K = 7	0.2824	0.2825	0.2844	0.3191	0.5202	0.5650	0.3553	
K = 8	0.1996	0.2681	0.2720	0.2808	0.3176	0.5467	0.5176	0.3548

**Table 4 entropy-21-00470-t004:** The Entropy Value of Samples Under Different Fault Conditions.

Motor Types	Label	SampEn I	SampEn II	SampEn III	SampEn IV	SampEn V	SampEn VI	SampEn VII	Desired Output
Normal	1	0.3241	0.4299	0.3757	0.2647	0.1455	0.2659	0.4664	1
		0.3091	0.4368	0.3818	0.2506	0.1338	0.2599	0.4445	1
		0.2914	0.4567	0.3811	0.2742	0.1630	0.2888	0.4837	1
Fault I	2	0.2239	0.5127	0.2319	0.1434	0.1818	-0.5116	0.3566	2
		0.2854	0.2787	0.2934	0.2787	0.2018	0.8924	0.4550	2
		0.3838	0.4191	0.4041	0.3033	0.1318	-0.8236	0.6149	2
Fault II	3	0.3084	0.3857	0.5754	0.2425	0.7845	0.3938	0.6215	3
		0.0684	0.5457	0.5754	0.2625	1.5645	0.4307	0.2915	3
		0.2384	0.5557	0.5130	0.2225	1.4085	0.4553	0.5515	3
Fault III	4	0.2749	0.6835	0.1855	0.1931	0.1958	0.3765	0.5568	4
		0.4348	0.3811	0.1955	0.5119	0.2458	0.3519	0.4968	4
		0.4840	0.5323	0.1755	0.2435	0.3758	0.3273	0.4568	4

**Table 5 entropy-21-00470-t005:** Final Recognition Results.

Classifier	Experimental Parameters	Recognition Rate				
		Normal	Fault I	Fault II	Fault III	All Test Samples
LQV	Hidden Layer = 5;	75%	75%	80%	80%	77.5%
	Hidden Layer = 20;	85%	90%	90%	90%	88.75%
	Hidden Layer = 40	80%	85%	85%	85%	83.75%
ELM	Activation Function (Sin);	80%	80%	85%	85%	82.5%
	Activation Function (Sig);	85%	90%	90%	90%	88.75%
	Activation Function (Hardlim)	80%	85%	85%	85%	84.5%
Decision Tree	Decision Tree (100);	85%	80%	80%	85%	83.75%
	Decision Tree (500)	95%	90%	85%	85%	88.75%
	Decision Tree (800)	90%	85%	80%	80%	83.5%
Random Forest	Decision Tree (100);	100%	90%	90%	90%	92.5%
	Decision Tree (500)	100%	95%	90%	90%	93.75%
	Decision Tree (800)	95%	90%	90%	90%	91.25%

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
