# Peer review of "Mechanical Fault Diagnosis of a DC Motor Utilizing United Variational Mode Decomposition, SampEn, and Random Forest-SPRINT Algorithm Classifiers"

_entropy, 2019, doi:10.3390/e21050470_

Round 1

Author Response

Thank you very much you're your letter and the referees' reports. Based on your comment and request, we have made extensive modification on the original manuscript. Here, we attached revised manuscript. In the formats of PDF, you're your approval. A document answering every question from the referees was also summarized and enclosed. A revised manuscript, with the correction section was attached as the supplemental material and for easy check purpose. Should you have any question, please contact us without hesitate 

Point 1: The reviewer’s comment: In abstract in VMD they define “V” as “Variational”. I suppose they mean Vibrational Mode.

Response 1: VMD is a new variable-scale signal processing method, in which V represents the meaning of variable-scale. It can decompose complex signals into K (AM-FM) component signals of preset scale. Because K can be preset, if K value is appropriate, the phenomenon of mode aliasing can be effectively suppressed. So Variation means finding the optimal solution of original signal decomposition by optimizing K value. The “V” also mean vibrational mode, but it not normal vibrational mode, it mean variable vibration modes. According to different K, the signal will be decomposed different mode.

Point 2: Author should give credit to literature and discuss these points in the paper.

Response 2: we agree with your comment, we rewrite the entropy section in page 2, line 71. Equation 14 is essentially probability.It is SampEn(m,r)= limN{-ln[Am(r)/Bm(r)]},special form, when N is a finite value. We rewriter equation 14 as SampEn(m,r,N)={-ln[Am(r)/Bm(r)]}

Reviewer 2 Report

Regarding the paper entitled "Mechanical Fault Diagnosis of DC Motor Utilizing 3 VMD United SampEn and Random  Forest-SPRINT Algorithm Classifier ", here are some minor comments:

Another important method for fault diagnosis based signals are based on the analysis of the principal component of the data, for instance, consider the following papers in the introduction:

https://doi.org/10.1016/j.ifacol.2018.09.604

https://doi.org/10.1016/j.conengprac.2018.09.006

*What are the criteria to choose the simulated signals given on page 4, line 155. 

* Why these signals are noise free?.

* Explain in more detail how to obtain Figure 2 and 3.

*Rewrite the algorithm given in Section 3 (Sample Entropy) as an Algorithm (in a table).

*As shown in Fig. 16,  Envelope Spectrum of Each Mode clearly show a fault signature. However, what kinds of faults are considered, how these faults are induced to the physical system, are these faults typical in this kind of systems. it is possible to consider also sensor faults?. 

 *The title says fault diagnosis, but I only see fault detection. 

Author Response

I am very grateful to your comment for the manuscript. According you're your advice, we amended the relevant part in manuscript. Some of your questions were answered below

Point 1: Another important method for fault diagnosis based signals are based on the analysis of the principal component of the data. I am very grateful to your comment for the manuscript. According you're your advice, we amended the relevant part in manuscript. Some of your questions were answered below

Response 1: We have discussed two methods in the paper which given on page 2, from line 54 to 70. Meanwhile, by discussing the two methods in detail, we consider the two methods are inappropriate for the paper.

Point 2: What are the criteria to choose the simulated signals given on page 4, line 155.

Response 2: Based on the literature WANG Hongchao, CHEN Jin, DONG Guangming. Fault Diagnosis Method for Rolling Bearing’S Weak Fault Based on Minimum Entropy Deconvolution and Sparse Decomposition[J]. JOURNAL OF MECHANICAL ENGINEERING, 2013, 49(1):88-94. The paper discuss the specific method how to construct simulation signal.

Point 3: Why these signals are noise free?

Response 3: These signals are not noise free, before decomposing signals, we have employed for signals denoising. We relist the noise signal figure in the paper(in page 14 ). Meanwhile, we have discussed the denoising method in the paper, comparing with traditional methods include Fourier transform method, time-frequency analysis method, optimal filtering or estimation method and adaptive filtering method, wavelet transform applied variable time and frequency windows to analyze signals. It is very suitable for non-stationary vibration signals. Based on the above statements, we decide to choose wavelet transform for denoising signals.

Point 4: Explain in more detail how to obtain Figure 2 and 3

Response 4: We will relist more detail in the page5. Figure 2 is the spectrum distribution of input signal, it represents the frequency of the three sub-components of the synthesized signal. Figure 3 is spectrum distribution of decomposed modal signals, from the comparison of Figure 2 and figure 3, we have seen that the three decomposed modes have a good agreement with the frequency of the original signal in the frequency range.

Point 5: Rewrite the algorithm given in Section 3 (Sample Entropy) as an Algorithm (in a table).

Response 5: We rewrite the section 3 (Sample Entropy) for pseudo code.

Point 6: As shown in Fig. 17, Envelope Spectrum of Each Mode clearly show a fault signature. However, what kinds of faults are considered, how these faults are induced to the physical system, are these faults typical in this kind of systems. It is possible to consider also sensor faults?

Response 6: Three typical faults are considered: rotor asymmetry load- Shafting misalignment and rotor system's misalignment-rubbing coupling faults. Density of envelope graph indicates that IMF component contains fault feature information, the more fault feature information the IMF contains, the smaller the envelope entropy is, the more dense the envelope graph is, hence, it can reflect its physical system from the fault information. Because the experiment mainly simulates three typical faults of motor, it will discharge interference from other aspects to motor fault signal, especially the sensor aspect will be strictly controlled. Therefore, sensor faults can be eliminated, and motor fault signals are all generated by itself.

Point 7: The title says fault diagnosis, but I only see fault detection.

Response 7: Fault detection: Real-time monitoring process data to determine whether a fault has occurred. Fault diagnosis: To judge which kind of fault occurs, what the fault type is and where location of the fault is. By definition, we can get the difference between fault diagnosis and fault analysis. In this paper(page 14), three types of faults which are not indicated are relisted, the three types of faults are respectively: rotor asymmetry load- Shafting misalignment and rotor system's misalignment-rubbing coupling faults. Through the analysis of different signals, different four kinds of signals are classified to identify which fault type the signals belong to.

Thank you for your suggestion, I have revised the corresponding position of this paper in accordance with your comments. Thank you again for the advice you have made on this paper.

Round 2

Reviewer 1 Report

The following statement should be attributed to Sosnovskiy and Sherbakov [29]  NOT M.Naderi.[30]

88 "Furthermore, M.Naderi made an attempt to propose a generalized theory of evolution based on the concept of tribo-fatigue entropy. Meanwhile, mechanothermody namical function is constructed for the specific case of fatigue damage of  materials due to the variation of temperature from 3 K to 0.8 of melting temperature based on the analysis of 136 experimental results[30]."

Naderi et al.  [30] proved  experimentally that cumulative entropy generation is constant at the time of failure and is independent of geometry, load and loading frequency. They named this critical entropy value Fatigue Fracture Entropy (FFE).

References are not in standard format. They should be written in standard reference format

Author Response

Thank you very much you’re your letter and the referees’ reports. Based on your comment and request, we have made extensive modification on the original manuscript. Should you have any question, please contact us without hesitate.

Point 1: The following statement should be attributed to Sonsnovskiy and Shenbakov[29] NOT M. Naderi[30]

Response 1: I have revised the citation of two literature.

Point 2: Reference are not in standard format. They should be written in standard reference format.

Response 2: We have written in standard reference format

Thank you for your suggestion, I have revised the corresponding position of this paper in accordance with your comments. Thank you again for the advice you have made on this paper.